# Replication study of "Explaining in Style: Training a GAN to explain a classifier in StyleSpace"

## Reproducibility Summary

**Scope of Reproducibility**

In this report claims made in the paper "Explaining in Style: Training a GAN to explain a classifier in StyleSpace" will be tested. This paper claims that by creating a generative model based on pre-trained classifier it is possible to discover and visually explain the underlying attributes that influence the classifier output which can lead to counterfactual explanations. From this it can be deduced what classifiers are learning.

**Methodology**

To reproduce the StylEx architecture that has been proposed an already existing implementation of the styleGAN2 model is modified. To implement the AttFind algorithm found in the paper the original TensorFlow code has been converted in to PyTorch code. Furthermore due to the restraint of only having access to one GPU the image resolution has been down scaled to 64x64 pixels such that computation time will not be to extensive.

**Results**

A model is created for both dog-cat and age classification. The models performed worse than stated than the pretrained models, most likely due to some issues with the StylEx style space, as AttFind performed well on the StyleGAN2 model. Due to the limitations in the adaptation it is not possible to definitively state whether the claims are true or false.

**What was easy**

Pretrained models and the AttFind algorithm were available for execution. It was therefore possible to quickly obtain some results given in the original paper by the authors. It gave a good baseline of what to expect should everything run correctly.

**What was difficult**

No training code or information on the training procedure was available publicly, meaning it had to be created from scratch. Although the AttFind algorithm was available, it was in TensorFlow and not PyTorch therefore this needed to be converted. Implementing and training everything ended up taking a lot of time and resources, causing a hyperparameter search and further research not to be possible.

**Communication with original authors**

We have had contact with the original authors and many of our questions about their paper were answered. Response time was fast as well, usually taking no longer than 40 hours.

# 1  Introduction

The task of classification is a common task in the field machine learning. The ability to recognize complex attributes and separate large quantities of data into categories makes deep models useful tools for this task. A disadvantage to using these deep classifiers can be that deep models are not easily explained, which makes it unclear why data is classified in a certain way. This is a problem because without a method to explain the classifier's decisions, it is not clear whether the model bases its decisions on valid attributes or on some bias.

One method to explain a deep classifier is to use counterfactual explanations.[7] [8] Here, decisions of the classifier can be explained by observing how changes in the input data influence the classifier output. If changing an attribute of some data point has some substantial influence on the classification of that data point, it can be learned that this attribute was important for the classification. In the paper "Explaining in Style: Training a GAN To Explain a Classifier in StyleSpace" [5] the authors expand on this idea by developing a method that can find and visualise the most important attributes for a specific classifier's decisions.

Lang et al. state that by using the "StyleGAN2" architecture [4], they can utilize the trait that this has a disentangled latent space [9] to extract individual attributes that are semantically interpretable. Furthermore by incorporating a fully trained classifier into the training process of the styleGAN2 architecture, this disentangled latent space can be manipulated to find the attributes that change the prediction of the classifier.

The following paper will be an analysis of the research performed and will asses its claims and its reproducibility.

# 2  Scope of reproducibility

The original paper proposes the StylEx model. This model is an adaptation of the StyleGAN2 model. It adds an encoder, which makes it so that counterfactuals of specific images can be generated, and a classifier, which makes it so that classifier specific observations can be made. The paper also proposes its AttFind algorithm, which is designed to find classifier-specific attributes in the trained models. The paper makes the following main claims about these implementations:

- The StylEx model is able to reconstruct images based on a specific classifier's output by incorporating this classifier in its training and model input.

- The AttFind algorithm can find important style space coordinates for the classifier, which can be used to generate counterfactual explanations where semantically interpretable attributes are changed in images to show their importance in the classifier's decision making.

- Their model is able to more effectively find features that more accurately explain the classifier, meaning that changing these features should allow the classification to flip more frequently than previous methods. The main comparison done is with the work of Wu *et al.*[9], where changing the 10 most important features proved much more effective on the StylEx method for all datasets used.

This paper will focus on these three claims by examining the extent to which they hold. For this both a pretrained model that was provided by the authors of the original paper, as well as some models that were trained from scratch will be researched.

# 3  Methodology

## 3.1  Models

### 3.1.1  Classifier

The MobileNetV2 architecture was used as the classifier. No pretrained models exist for the classification task at hand, thus an untrained model was taken and trained on the chosen datasets.

### 3.1.2  Encoder and Discriminator

The discriminator architecture is defined in the styleGAN2 paper [4]. The encoder and decoder have the same architecture, the architecture of these two models is therefore a residual discriminator without progressive growing. The

only difference between the two architectures is the final linear layer, where for the encoder the output is mapped to an encoding dimension of 512 and for the encoder the output is mapped to a single value. Even though the encoder and discriminator share almost the same architecture their functionality and training is different. The encoder is utilized to encode an image into a latent vector to input into the generator model whereas the discriminator is used to classify whether an image is generated by the generator or is a real image.

### 3.1.3 Generator

For the generator a modified version of the StyleGAN2 architecture was used. Figure 1 illustrates this architecture. As its input it takes the encoded latent vector of some image concatenated with the classifier output on this image. This is then mapped to the style space by multiple affine transformations. These style vectors can then be input to the synthesis network, which uses a multitude of convolutional layers and skip connections to generate an image. The original StyleGan2 architecture also utilised a mapping network that mapped some noise vector z to the latent vector. Contact with the authors of the original paper revealed that the reported results in their paper were achieved by alternating between using the encoder and using this mapping network during training. However, since this was not mentioned in the paper and since the authors advised that only using the encoder would also give good results and would lead to a faster convergence, the decision was made to only use the encoder to obtain the latent vector in this research.

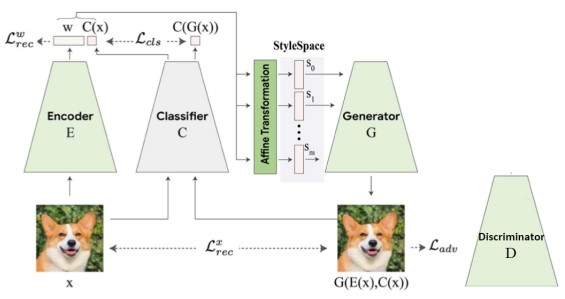

Figure 1: The StylEx model proposed by Lang *et al.* [5]

### 3.2 Training the StylEx model

During the training process of the StylEx model, the encoder, discriminator, and generator are all trained simultaneously. At each iteration the encoder and classifier get a batch of training images. The output of both these models are concatenated and used by the generator network to reconstruct the original images. To calculate the loss there are four main loss terms, namely the adversarial loss, the path regularization loss, the reconstruction loss and the classifier loss.

The adversarial loss is a general loss term for GANs over the outputted image from the generator inputted into the discriminator [2]. The path regularization loss causes the latent vectors w to be regularized based on current and previous latent vectors such that the path length does not diverge from the mean path lengths leading to more consistently behaving models [4]. The reconstruction loss is made up of an *Learned Perceptual Image Patch Similarity*(LPIPS) distance between the real image and the generated image [10], an L1 loss between the image and the generated image and an L1 loss between the the output of the encoder on the real image and the generated image. From further communication with the authors it was found that the LPIPS loss and L1 loss between the image and the generated image have a weight of 0.1. Lastly, the classifier loss is the *Kullback–Leibler*(KL) divergence between the classifier output on the original image and the generated image.

### 3.3 AttFind

The AttFind algorithm is an algorithm that will try to uncover classifier specific attributes. The input of AttFind is the classifier, the generator, the threshold, and a set of images whose predicted label by the classifier differs from the label of the images that are to be generated. For every image the algorithm will iterate through the style coordinates and apply a different direction for every style coordinate on the image and calculate its effect on the classifier. The coordinate with the largest effect on the classifier output over the set of images is selected. All images on which this style coordinate has a large effect will be removed from the images list and the style coordinate will be put in a list. Finally, when all the images are removed from the images list or if all the style coordinates have resulted in a large change in the classifier the output of the algorithm will be the style coordinates that had a large effect on the classifier and the direction in which they where changed. With these coordinates and directions of each feature new images can be generated where the effect of changing these coordinates shows a difference in the image and the classification.

### 3.4 Datasets

The datasets used for the experiments are shown in Table 1. Both the StylEx and classifier datasets match those used in the original paper. Due to computational limitations all images were scaled down to 64x64.

| Feature | StylEx Dataset | Classifier Dataset |
|---|---|---|
| Cats/Dogs | AFHQ (9895 train, 1003 validation)[1] | AFHQ (9895 train, 1003 validation) |
| Age | FFHQ (60000 train, 10000 validation) [3] | CelebA (71968 train, 10073 validation) [6] |

Table 1: Datasets used with the amount of training and validation images per dataset.

### 3.5 Hyperparameters

Although they were not included in the paper itself the authors responded quickly and supplied the hyperparameter details. The learning rate of the original model was set to 0.002 and the batch size was set to 16. The authors had 8 GPUs at their disposal and therefore the batch size per GPU was 2. Furthermore the original experiment was run for 250k iterations. During training for this paper however, problems were encountered using these hyperparameter settings due to the decreased resolution of the images and GPU limitations. Therefore the batch size was decreased to 4 and the learning rate was decreased by a factor 10 to 0.0002 accordingly. Furthermore, 220k iterations were run, which took approximately 48 hours. This was chosen to be slightly lower than the amount of iterations run in the original paper, both because of time and resource constraints and because of the lower resolution images in this research leading to an observably faster convergence.

### 3.6 Experimental setup and code

#### 3.6.1 Available resources

The available resources for the experiments are limited. Although the AttFind algorithm with some pretrained models is publicly available [1], the current implementation is in TensorFlow meaning it had to be adapted into PyTorch first. Aside from this no training code is available, and the pretrained models do not offer many insight towards the training procedure. This means the StylEx model and training code had to be implemented again based on the details provided by the paper and the authors. This self implementation was done by adapting an existing PyTorch implementation[2] of StyleGAN2 into the StylEx model. The full code and other resources are available on GitHub [3].

#### 3.6.2 Image reconstruction

As stated before, the original paper claimed the StylEx model is able to reconstruct images based on the classifiers output. This claim is verified through visual representation and will thus be done the same in this review. Analyzing random selections of image generations compared to their original should give a quick overview of how effective these reconstructions are.

#### 3.6.3 Semantically interpretable features

The original paper claims that the AttFind algorithm is able to generate counterfactual explanations of images, which describe semantically interpretable attributes. If the counterfactual explanations are semantically interpretable attributes, these would have to be noticeably different from one another. To test this first the top 4 features are extracted. The user study from the original paper is then remade. At each section this user study contains two GIFs of the same feature changing in different images. The users then have to choose from two different GIFs of changing images which one has the same change in feature as the first two. The user study was shared with personal connections from various backgrounds.

---

[1]https://github.com/google/explaining-in-style
[2]https://github.com/rosinality/stylegan2-pytorch
[3]https://anonymous.4open.science/r/ExplainingInStyleReproduce-8ECE/README.md

### 3.6.4 Flipping the prediction

Finally in order to test the validity of the claim that the StylEx model is more effectively able to flip the classifier prediction than other models such as the one by Wu *et al.*, the experiment from the original paper is recreated. The Wu *et al.* algorithm works by considering the normalized differences of style space values of images in one classifier class with the mean of all images. Wu *et al.* ensure images that strongly exhibit one class by selecting the top 2% of images that conform to the class the most. Due to limited computing resources this would result in few examples, so a classifier logit threshold of 0.9 is used instead. Wu *et al.* also do originally consider the direction of change necessary for the desired classifier effect. For fair comparison, These directions are obtained by examining whether the mean of the differences is positive or negative. It is not known whether Lang *et al.* do the same. To recreate the original experiment the latent vectors are used to generate images that have their top $k$ features flipped, in this case 10 features. The classification is then compared against the original image, where an image will count as flipped if it is now classified as another class. The results are calculated as the total percentage of images where classification flipped after the attributes were changed within the style space.

### 3.7 Computational requirements

The StylEx model, which include the encoder, generator and discriminator, were trained on one GeForce 1080 TI GPU. The total runtime for the training was 48 hours using a batch size of 4. The cat/dog classifier model was trained on the Google Colab GPU, an NVIDIA Tesla K80 GPU, for approximately 30 minutes. The age classifier model was trained on one GeForce 1080 TI GPU for approximately 6 hours.

## 4 Results

The results of the experiments will be shown and discussed in the following sections. In section 4.1 the overall reconstruction will be analysed, in order to determine how effective the reconstructions have been. Next in section 4.2 the findings of the style space coordinates will be discussed together with the results of the user study, to test how semantically interpretable the found features truly are. Finally in section 4.3 the results of the feature change on the classifier output will be analysed and shown.

The images shown were randomly selected among the images belonging to the required task according to the classifier.

### 4.1 Image reconstruction

When looking at the visual results of the reconstruction of the images by the AttFind algorithm which are shown in Figure 2, it can be seen that there is some clear reconstruction, but that the model does not recreate the images perfectly using this lower resolution data. One issue to note is how the model handles younger animals. When looking at the second column in Figure 2, it can clearly be seen that this is reconstructed as a more adult version of the dog instead of the puppy it originally was. This could be a limitation of the StylEx style space, or it could be due to the amount of available training data on younger animals. Some other animals appear to be changing features all together. When observing the changes in the third column of Figure 2, the generated cat only seems to share its colour with the original cat, as well as the overall pose. Aside from these two features the images are completely different cats. Results for the FFHQ model were similar.

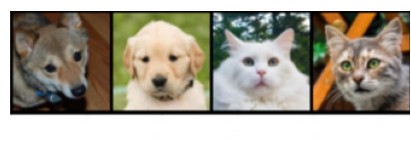
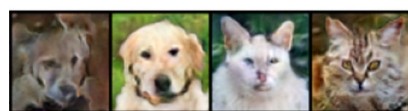

Figure 2: AFHQ reconstruction results. Top row shows the original images. Bottom row shows the reconstructed images by the model. The images have a lot of their elements changed in the style space.

### 4.2 Semantically interpretable features

#### 4.2.1 Features for the pretrained model

In Figure 3, qualitative results of the pretrained model architecture are shown. The features shown are clearly semantically interpretable, and change the classification score significantly.

The top 4 features found for the pretrained model on the FFHQ by the AttFind algorithm were: skin complexity, eyebrow thickness, glasses, and hair colour. This matches the results reported in the original paper.

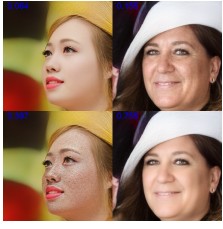

(a) Skin complexity

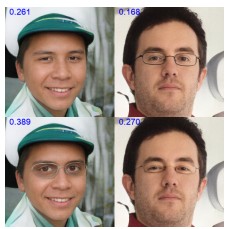

(b) Glasses

Figure 3: Results of the AttFind algorithm with the pretrained models. (a) shows the effect of skin complexity, and (b) shows the effect of having glasses on image classification. For every image the original generated images are the top two images with their classifier score belonging to the other class in blue and the bottom images are the images with changed features and the belonging classification score.

### 4.2.2 Features for the new models

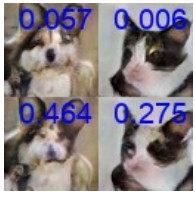

(a) First feature

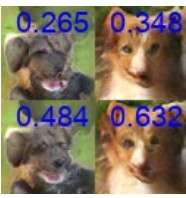

(b) Second feature

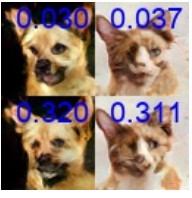

(c) Third feature

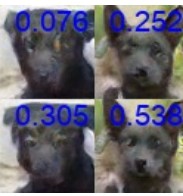

(d) Fourth feature

Figure 4: Four most important features found for StylEx AFHQ, with the most important feature on the left and the fourth most important feature on the right. The images of the dogs show the classification score as cats, the cats show the classification score for being classified as dogs.

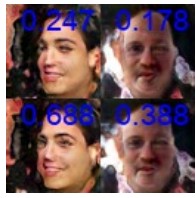

(a) First feature

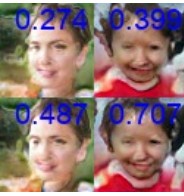

(b) Second feature

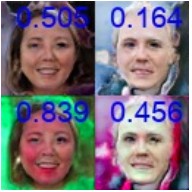

(c) Skin colour

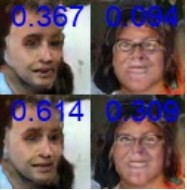

(d) Open mouth

Figure 5: Four most important features found for StylEx FFHQ, with the most important feature on the left and the fourth most important feature on the right. Left column shows classification as old, right column shows the young classification

Running the same experiment for the StylEx model gave the following results as shown in Fig. 4. In these images it is much less clear what distinct features exist for the low resolution AFHQ data. Not only are the features barely distinguishable, but the changes that are visible do not necessarily apply to any real world semantics. This could possibly be due to the complexity of the data, as the model trained on AFHQ data set was found to perform less well than models trained on human data in the original paper as well. Alternatively, this could be because of an issue in either the AttFind algorithm or the StylEx style space.

As can be seen in Figure 5, the results for the FFHQ data is very similar to the results for the AFHQ data, although a bit better. The first two features found by AttFind do not seem to be connected to any semantically interpretable features. Looking at the third most important feature however there seems to be some form of change in skin colour, although the change itself is not exactly realistic. The final feature seems to reference back to the opening of the mouth, which is the

most semantically interpretable feature of the four. These results therefore somewhat confirm the hypotheses made above, since the model indeed seems to perform slightly better on the FFHQ data set, but still does not validate the claim that semantically interpretable features are found.

### 4.2.3 Features for the StyleGAN2 model

To research the cause of the slightly disappointing results of the trained StylEx models, an investigation of the original StyleGAN2 model could give some more insight. Figure 6 shows the results of the AttFind algorithm on a StyleGAN2 model trained on the AFHQ data and a StyleGAN2 model trained on the FFHQ data with the same hyperparameters as the models in the previous section. Note that these images are not counterfactuals of specific images in the validation data as before, but rather they are counterfactuals of images generated from some randomly generated z vector, since the original StyleGAN2 architecture does not include an encoder. As can be seen, changing the attributes found by AttFind does give some different results here than in the previous section. The changed attributes seem to be semantically impactful, since clear changes in respectively facial structure, coat colour, glasses, and skin colour can be seen. From this it could be concluded that the problem in the previous section is not the AttFind algorithm nor is it the image resolution, since both are the same between these two sections. Therefore it would be likely that the problem lies somewhere in the trained StylEx model. The most likely theory for this is that our implementation does not use these randomly sampled z vectors that the StyleGAN2 model does use. Therefore it could be the case that without this random sampling the model only gets similar images, namely only the training data, which could result in a less defined style space and thus in that less interpretable features are found.

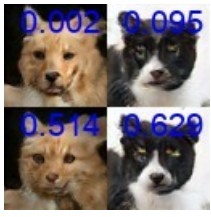 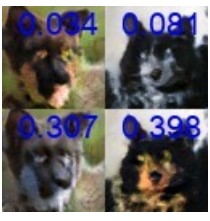 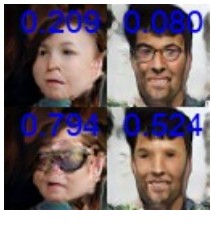 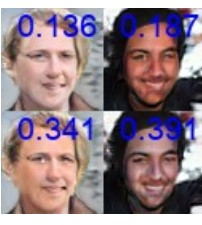

(a) facial features        (b) coat colour        (c) glasses        (d) skin colour

Figure 6: The two most significant features found for the StyleGAN2 model on the AFHQ data (6a, 6b) and the FFHQ data (6c, 6d).

### 4.2.4 User study

In total 61 responses were recorded in the user study. The original paper achieved an accuracy of 0.983 (±0.037) on an unknown amount of questions and participants. The results of the new poll include 3 questions for each feature in the top 6 features found by AttFind for a total of 18 questions. The overall score of this poll was lower, as it achieved a score of 0.918 (±0.038), possibly due to a difference in demographic. The user study still shows an overall good understanding of the features, as a score of over 90% was achieved. The first question got the worst results, possibly due to people not fully understanding how the study worked or due to the more subtle feature (skin complexity) shown.

## 4.3 Flipping the prediction

|  |  | Wu *et al.* | Attfind with StyleGAN2 | AttFind with StylEx |
|---|---|---|---|---|
| New | AFHQ | 0.247 (± 0.012) | 0.391(± 0.022) | 0.042 (± 0.006) |
|  | FFHQ | 0.253 (± 0.015) | 0.678 (± 0.016) | 0.429 (± 0.013) |
| Original | AFHQ | 0.010 | - | 0.250 |
|  | FFHQ | 0.169 | - | 0.939 |

Table 2: Flip percentage using the 64x64 images, as well as the original results for the datasets.

Table 2 shows the results of running the Wu *et al.* algorithm as well as the StylEx method with AttFind. The new results are much lower than those found in the original paper. As mentioned before, our most likely hypothesis for this is that this is because of the lack of z mapping performed, resulting in a lesser style space. The model had more trouble with

the AFHQ classification than the FFHQ, which does fall in line with the original results. This is probably due to the fact that cats and dogs are far more binary than age and therefore when z mapping is not performed the model does not obtain enough varied inputs. An interesting point to note is that the results of the Wu *et al.* paper are much higher than reported in the original report. Especially the AFHQ results are of note here, as only a 1.0% flip rate was achieved originally, but the StyleGAN2 model with the reduced resolution achieves a flip rate of 20.8%.

# 5 Discussion

## 5.1 Conclusions

From section 4, the different claims stated in section 2 can be supported or contradicted. The first claim states that the StylEx model is able to reconstruct images based on a specific classifier. In the results some clear reconstruction could be seen although the images still had a lot of problems. This concludes that with the available resources and information the StylEx model is able to reconstruct images based on specific classifier but not to the same complexity as stated in the paper.

Furthermore, the second claim states that the AttFind algorithm can be used to find the most important attributes that explain the classifier. From the results this claim seems to be supported. This is mostly because although the results from the AttFind algorithm on the stylEx model are suboptimal, the results on the StyleGAN2 model are feasible. From this it can be concluded that the AttFind algorithm performs as expected and the attributes it finds could be used to generate counterfactuals.

Lastly, the third claim states that the StylEx model can more accurately explain the classifier than previous methods. In the results it was found that this was not the same for this implementation. The flip rate of the StylEx model trained on age classification was better than the wu *et al.* results but not better than the StyleGAN2 results and the StylEx model on the AFHQ data was the worst performing. From this it can be concluded that given the resources and time it was not possible to reproduce these results.

## 5.2 What was easy

The authors of the original paper made the AttFind algorithm as well as the pretrained models publicly available. This allowed results to be effectively extracted from them, and also allowed to easily validate some of the claims made in the paper and gave an overall good baseline for the experimentation. The results obtained were also mostly in line with what the paper reported. With the AttFind algorithm it was also possible to effectively obtain the results on newly trained models.

## 5.3 What was difficult

Since no training code was made publicly available by the authors this needed to be implemented from scratch in PyTorch, which took a significant amount of resources to complete. Although the AttFind model was publicly available, documentation itself was very limited, meaning that translating it from TensorFlow to PyTorch was a nontrivial task as well. These bottlenecks ended up affecting the amount of experiments that could be performed, and limited the opportunity to expand upon the paper as well. Another bottleneck that affected the experimentation of the report is the available resources. Only having access to Google Colab (which limits GPU usage) and a single GeForce 1080TI GPU limited the amount of time to run experiments, with training taking up a large portion of available GPU usage. This also meant the image quality had to be scaled down in order to effectively train the model, although this negatively impacted the quality of the obtained results. In the original paper the authors trained each model on 8 computationally stronger GPUs, which resulted in this difference in overall image reconstruction quality and resolution. After this little room was left to do things like hyperparameter search or further research given these constraints, which would have added to this review.

## 5.4 Communication with original authors

There was communication with the original authors about the internal structure of the models, as well as the hyperparameters which were not included in the original paper. They also answered any additional questions about the latent vectors and the use of lower dimension images for the model. It was also recommended to not use the z mapping if time was limited too much.

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

 # Appendix

 ## A    StyleGAN2 training results

 Displayed in this section are the results of the StyleGAN2 model after training, for reference to the results StylEx
 achieved.

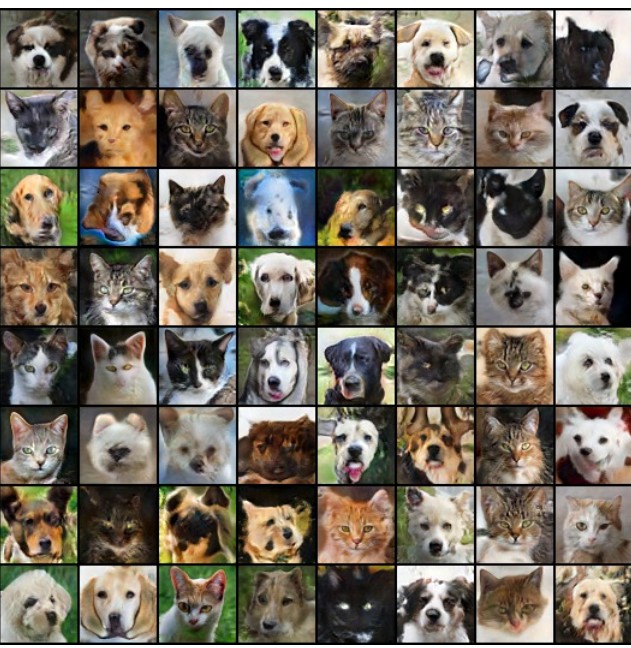

Figure 7: Generated results from the StyleGAN2 on the AFHQ dataset.

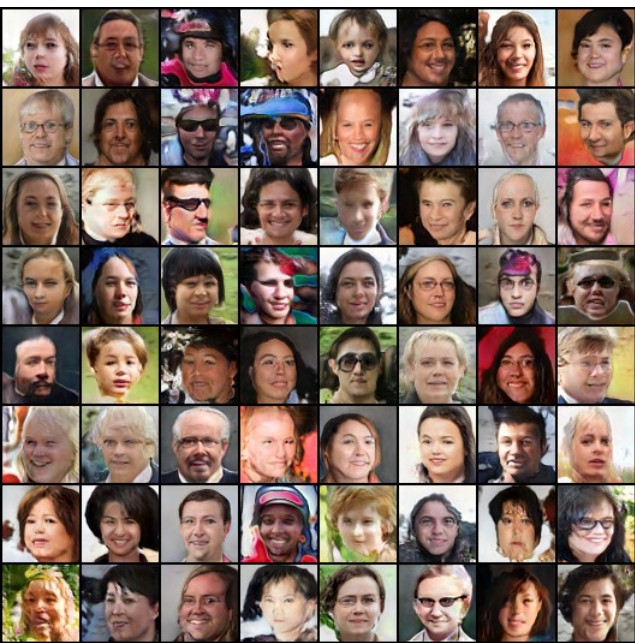

Figure 8: Generated results from the StyleGAN2 on the FFHQ dataset.

## B Wu *et al.* results

Below are some results of extracting the most important features of the Wu *et al.* paper.

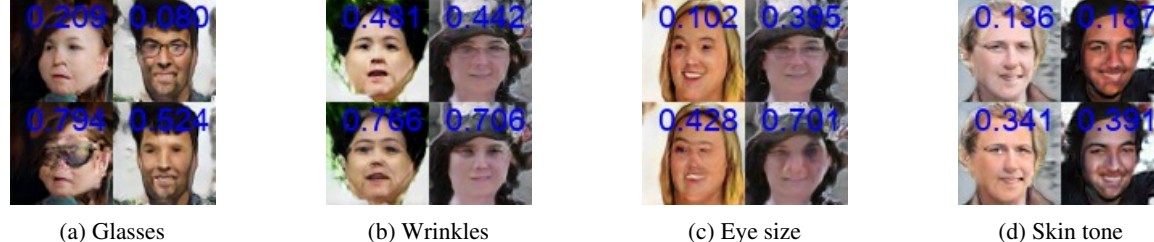

(a) Glasses            (b) Wrinkles            (c) Eye size            (d) Skin tone

Figure 9: Four most important features found for Wu *et al.*, with the most important feature on the left and the fourth most important feature on the right. Left image at each feature shows the classification score for being classified as old, the right shows the classification score for being classified as young.

## C Model architectures

More detailed architecture of the styleGAN2 model used in the paper.

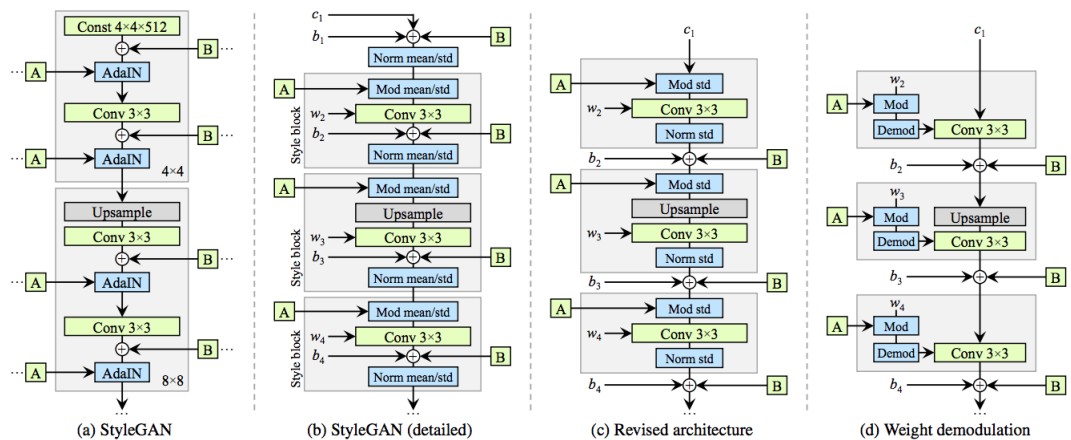

(a) StyleGAN        (b) StyleGAN (detailed)        (c) Revised architecture        (d) Weight demodulation

Figure 10: Original StyleGAN architecture (a and b) alongside the improved StyleGAN2 architecture (c and d), as shown in [4]. A is a learned affine transform from the latent code and B is some form of noise broadcast operation.

The architecture of the MobileNetV2 model. Used for the classifiers.

| Input | Operator | $t$ | $c$ | $n$ | $s$ |
|---|---|---|---|---|---|
| $224^2 \times 3$ | conv2d | - | 32 | 1 | 2 |
| $112^2 \times 32$ | bottleneck | 1 | 16 | 1 | 1 |
| $112^2 \times 16$ | bottleneck | 6 | 24 | 2 | 2 |
| $56^2 \times 24$ | bottleneck | 6 | 32 | 3 | 2 |
| $28^2 \times 32$ | bottleneck | 6 | 64 | 4 | 2 |
| $14^2 \times 64$ | bottleneck | 6 | 96 | 3 | 1 |
| $14^2 \times 96$ | bottleneck | 6 | 160 | 3 | 2 |
| $7^2 \times 160$ | bottleneck | 6 | 320 | 1 | 1 |
| $7^2 \times 320$ | conv2d 1x1 | - | 1280 | 1 | 1 |
| $7^2 \times 1280$ | avgpool 7x7 | - | - | 1 | - |
| $1 \times 1 \times 1280$ | conv2d 1x1 | - | k | - | |

Figure 11: MobileNetV2 architecture.

