# OpenReview forum: "Replication study of "Explaining in Style: Training a GAN to explain a classifier in StyleSpace""
_ML_Reproducibility_Challenge/2021/Fall — RC2021_

### Official Review · Reviewer_vHtx · 2022-02-28
**Solid reproducibility effort**

**Rating:** 6
**Confidence:** 4

**Review:**

**Reproducibility summary:** The summary is overall clear and well summarizes the authors' reproducibility effort.

**Scope of reproducibility:** The authors check the following three claims made by the original paper: 1) The StylEx model can reconstruct images based on a specific classifier’s output. 2) The AttFind algorithm can find important style space coordinates for the classifier, which can then be used to generate counterfactual explanations. 3) The StylEx model can more effectively find features that accurate explain the classifier.

**Code:** The authors convert the original author’s Tensorflow implementation of the AttFind algorithm into PyTorch. The authors had to implement the StylEx model and training code on it based on other existing implementations.

**Communication with the original authors**: The authors have corresponded with the original authors several times via email to receive additional details about the network architecture and hyperparameters. They note that the response time was fast.

**Hyperparameter search:** The authors note that they received hyperparameter details when they reached out to the authors. However, they had to make changes to the hyperparameters due to computational resource constraints. They note that implementing and training everything ended up taking a lot of time and resources, causing hyperparameter search and further research infeasible.

**Ablation study**: The authors did not conduct any ablation studies.

**Discussion on results:** The authors state that it is not possible to definitely state whether the original paper’s claims are true or false, due to the limitations in their adaptation and experiments. Section 4 that shows qualitative results with the pretrained model, the StylEx models, and and the StyleGAN2 models was helpful.

**Recommendations for reproducibility:** The authors don't provide explicit recommendations to the original authors. However, their descriptions of the difficulties they ran into may help the original authors improve the reproducibility of the work and/or future researchers building on this work

**Results beyond the paper**: There are no results beyond the original paper.

**Overall organization and clarity:** The report was organized, but the writing could be more clear.

---

### Official Review · Reviewer_w9bg · 2022-03-08
**Well-written and thorough report offering insights about the extent to which claims are supported in constrained settings**

**Rating:** 8
**Confidence:** 4

**Review:**

The authors attempt to replicate the study conducted in the paper titled — “Explaining in Style: Training a GAN to explain a classifier in StyleSpace”. As mentioned by the authors, the primary claim of the original paper is that by training a generative model based on a pre-trained classifier it is possible to discover and explain the underlying attributes that affect the output of the classifier. The authors of the original paper released their code in Tensorflow and the authors of this report attempted to reproduce the same in PyTorch. Additionally, due to computational prohibitive constraints, the report authors had to adopt downsampling and other strategies that likely makes their implementation slightly different when considered from an overall perspective. The authors were not able to conclusively justify the claims made in the original paper and hypothesize that it’s likely due to differences in adaptation.

**Strengths**

1. The report is generally well-written and easy to follow. The authors do a good job of outlining the overall takeaways from their reproducibility experiments in the “Reproducibility Summary” presented on the first page. Additionally, the authors do a good job of (1) outlining a preamble of the proposed approach (L29-43), (2) outlining the claims being investigated to the extent possible, and (3) the underlying methodology and the architectural details. All of this helps the reader get a quick grasp of the preliminaries involved in the original paper.
2. The authors very clearly highlight the experimental settings involved in the report — details surrounding the choice of datasets and associated training and validation splits (Section 3.4), how hyper-parameters were chosen keeping computational constraints in mind (Section 3.5), what resources they had access to off-the-shelf (L134-139) and other experimental details surrounding several ablations (for instance, Section 3.6.4).
3. I particularly appreciate how the authors attempt to conduct a smaller-scale version of the human study for the generated counterfactual explanations. Although not exactly reflective of the user study conducted in the original paper, the trends generally seem to support the claim of “overall good understanding of features” (Section 4.2.4).
4. Overall, I think the report is generally well-written and the conducted thorough experiments and drawn insights help understand the extent to which claims from the original paper hold when considered in limited capacity settings as considered in this report.

I don’t have major weaknesses to point out about the report.

---

### Meta-Review · Program_Chairs · 2022-04-09

**Recommendation:** Accept
**Confidence:** 5

**Metareview:**

A solid contribution to the reproducibility challenge.  The submission is accepted.

---

### Decision · Program_Chairs · 2022-04-09

**Decision:**

Accept

**Comment:**

Following the recommendation of reviewers and meta-reviewer, the paper is accepted for ML Reproducibility Challenge 2021, and will be published in the upcoming special edition of ReScience Journal.